# Memory Augmented Policy Optimization for Program Synthesis and Semantic Parsing

**Chen Liang**
Google Brain
crazydonkey200@gmail.com

**Mohammad Norouzi**
Google Brain
mnorouzi@google.com

**Jonathan Berant**
Tel-Aviv University, AI2
joberant@cs.tau.ac.il

**Quoc Le**
Google Brain
qvl@google.com

**Ni Lao**
SayMosaic Inc.
ni.lao@mosaix.ai

## Abstract

We present Memory Augmented Policy Optimization (MAPO), a simple and novel way to leverage a memory buffer of promising trajectories to reduce the variance of policy gradient estimates. MAPO is applicable to deterministic environments with discrete actions, such as structured prediction and combinatorial optimization. Our key idea is to express the expected return objective as a weighted sum of two terms: an expectation over the high-reward trajectories inside a memory buffer, and a separate expectation over trajectories outside of the buffer. To design an efficient algorithm based on this idea, we propose: (1) memory weight clipping to accelerate and stabilize training; (2) systematic exploration to discover high-reward trajectories; (3) distributed sampling from inside and outside of the memory buffer to speed up training. MAPO improves the sample efficiency and robustness of policy gradient, especially on tasks with sparse rewards. We evaluate MAPO on *weakly supervised* program synthesis from *natural language* (semantic parsing). On the WIKITABLEQUESTIONS benchmark, we improve the state-of-the-art by $2.6\%$, achieving an accuracy of $46.3\%$. On the WIKISQL benchmark, MAPO achieves an accuracy of $74.9\%$ with only weak supervision, outperforming several strong baselines with full supervision. Our source code is available at goo.gl/TXBp4e.

## 1 Introduction

There has been a recent surge of interest in applying policy gradient methods to various application domains including program synthesis [26, 17, 68, 10], dialogue generation [25, 11], architecture search [69, 71], game [53, 31] and continuous control [44, 50]. Simple policy gradient methods like REINFORCE [58] use Monte Carlo samples from the current policy to perform an *on-policy* optimization of the expected return objective. This often leads to unstable learning dynamics and poor sample efficiency, sometimes even underperforming random search [30].

The difficulty of gradient based policy optimization stems from a few sources: (1) policy gradient estimates have a large *variance*; (2) samples from a randomly initialized policy often attain small rewards, resulting in a slow training progress in the initial phase (cold start); (3) random policy samples do not explore the search space efficiently and systematically. These issues can be especially prohibitive in applications such as program synthesis and robotics [4] where the search space is large and the rewards are delayed and sparse. In such domains, a high reward is only achieved after a long sequence of *correct* actions. For instance, in program synthesis, only a few programs in the large combinatorial space of programs may correspond to the correct functional form. Unfortunately, relying on policy samples to explore the space often leads to forgetting a high reward trajectory unless it is re-sampled frequently [26, 3].

Learning through reflection on past experiences ("experience replay") is a promising direction to improve data efficiency and learning stability. It has recently been widely adopted in various deep RL algorithms, but its theoretical analysis and empirical comparison are still lacking. As a result, defining the optimal strategy for prioritizing and sampling from past experiences remain an open question. There has been various attempts to incorporate off-policy samples within the policy gradient framework to improve the sample efficiency of the REINFORCE and actor-critic algorithms (*e.g.*, [12, 57, 51, 15]). Most of these approaches utilize samples from an old policy through (truncated) importance sampling to obtain a low variance, but *biased* estimate of the gradients. Previous work has aimed to incorporate a replay buffer into policy gradient in the general RL setting of stochastic dynamics and possibly continuous actions. By contrast, we focus on deterministic environments with discrete actions and develop an *unbiased* policy gradient estimator with low variance (Figure 1).

This paper presents MAPO: a simple and novel way to incorporate a memory buffer of promising trajectories within the policy gradient framework. We express the expected return objective as a weighted sum of an expectation over the trajectories inside the memory buffer and a separate expectation over unknown trajectories outside of the buffer. The gradient estimates are unbiased and attain lower variance. Because high-reward trajectories remain in the memory, it is not possible to forget them. To make an efficient algorithm for MAPO, we propose 3 techniques: (1) memory weight clipping to accelerate and stabilize training; (2) systematic exploration of the search space to efficiently discover the high-reward trajectories; (3) distributed sampling from inside and outside of the memory buffer to scale up training;

We assess the effectiveness of MAPO on *weakly supervised* program synthesis from *natural language* (see Section 2). Program synthesis presents a unique opportunity to study *generalization* in the context of policy optimization, besides being an important real world application. On the challenging WIKITABLEQUESTIONS [39] benchmark, MAPO achieves an accuracy of $46.3\%$ on the test set, significantly outperforming the previous state-of-the-art of $43.7\%$ [67]. Interestingly, on the WIKISQL [68] benchmark, MAPO achieves an accuracy of $74.9\%$ without the supervision of gold programs, outperforming several strong *fully supervised* baselines.

## 2 The Problem of Weakly Supervised Contextual Program Synthesis

Consider the problem of learning to map a natural language question $\mathbf{x}$ to a structured query $\mathbf{a}$ in a programming language such as SQL (*e.g.*, [68]), or converting a textual problem description into a piece of source code as in programming competitions (*e.g.*, [5]). We call these problems *contextual program synthesis* and aim at tackling them in a weakly supervised setting – i.e., no correct action sequence $\mathbf{a}$, which corresponds to a gold program, is given as part of the training data, and training needs to solve the hard problem of exploring a large program space.

| Year | Venue | Position | Event | Time |
|---|---|---|---|---|
| 2001 | Hungary | 2nd | 400m | 47.12 |
| 2003 | Finland | 1st | 400m | 46.69 |
| 2005 | Germany | 11th | 400m | 46.62 |
| 2007 | Thailand | 1st | relay | 182.05 |
| 2008 | China | 7th | relay | 180.32 |

Table 1: $\mathbf{x}$: Where did the last 1st place finish occur? $\mathbf{y}$: Thailand

Table 1 shows an example question-answer pair. The model needs to first discover the programs that can generate the correct answer in a given context, and then learn to generalize to new contexts.

We formulate the problem of *weakly supervised* contextual program synthesis as follows: to generate a program by using a parametric function, $\hat{\mathbf{a}} = f(\mathbf{x}; \theta)$, where $\theta$ denotes the model parameters. The quality of a program $\hat{\mathbf{a}}$ is measured by a scoring or *reward* function $R(\hat{\mathbf{a}} \mid \mathbf{x}, \mathbf{y})$. The reward function may evaluate a program by executing it on a real environment and comparing the output against the correct answer. For example, it is natural to define a binary reward that is 1 when the output equals the answer and 0 otherwise. We assume that the context $\mathbf{x}$ includes both a natural language input and an environment, for example an interpreter or a database, on which the program will be executed. Given a dataset of context-answer pairs, $\{(\mathbf{x}_i, \mathbf{y}_i)\}_{i=1}^{N}$, the goal is to find optimal parameters $\theta^*$ that parameterize a mapping of $\mathbf{x} \to \mathbf{a}$ with maximum empirical return on a *heldout test set*.

One can think of the problem of contextual program synthesis as an instance of *reinforcement learning (RL)* with *sparse terminal rewards* and *deterministic transitions*, for which *generalization* plays a key role. There has been some recent attempts in the RL community to study generalization to unseen initial conditions (*e.g.* [45, 35]). However, most prior work aims to maximize empirical return on the training environment [6, 9]. The problem of contextual program synthesis presents a natural application of RL for which generalization is the main concern.

# 3 Optimization of Expected Return via Policy Gradients

To learn a mapping of (context $\mathbf{x}$) $\rightarrow$ (program $\mathbf{a}$), we optimize the parameters of a conditional distribution $\pi_\theta(\mathbf{a} \mid \mathbf{x})$ that assigns a probability to each program given the context. That is, $\pi_\theta$ is a distribution over the *countable* set of all possible programs, denoted $\mathcal{A}$. Thus $\forall \mathbf{a} \in \mathcal{A}: \pi_\theta(\mathbf{a} \mid \mathbf{x}) \geqslant 0$ and $\sum_{\mathbf{a} \in \mathcal{A}} \pi_\theta(\mathbf{a} \mid \mathbf{x}) = 1$. Then, to synthesize a program for a novel context, one finds the most likely program under the distribution $\pi_\theta$ via exact or approximate inference $\hat{\mathbf{a}} \approx \operatorname{argmax}_{\mathbf{a} \in \mathcal{A}} \pi_\theta(\mathbf{a} \mid \mathbf{x})$.

*Autoregressive* models present a tractable family of distributions that estimates the probability of a sequence of tokens, one token at a time, often from left to right. To handle variable sequence length, one includes a special *end-of-sequence* token at the end of the sequences. We express the probability of a program $\mathbf{a}$ given $\mathbf{x}$ as $\pi_\theta(\mathbf{a} \mid \mathbf{x}) \equiv \prod_{i=t}^{|\mathbf{a}|} \pi_\theta(a_t \mid \mathbf{a}_{<t}, \mathbf{x})$, where $\mathbf{a}_{<t} \equiv (a_1, \ldots, a_{t-1})$ denotes a prefix of the program $\mathbf{a}$. One often uses a recurrent neural network (*e.g.* [20]) to predict the probability of each token given the prefix and the context.

In the absence of ground truth programs, policy gradient techniques present a way to optimize the parameters of a stochastic policy $\pi_\theta$ via optimization of *expected return*. Given a training dataset of context-answer pairs, $\{(\mathbf{x}_i, \mathbf{y}_i)\}_{i=1}^N$, the objective is expressed as $\mathbb{E}_{\mathbf{a} \sim \pi_\theta(\mathbf{a} \mid \mathbf{x})} R(\mathbf{a} \mid \mathbf{x}, \mathbf{y})$. The reward function $R(\mathbf{a} \mid \mathbf{x}, \mathbf{y})$ evaluates a complete program $\mathbf{a}$, based on the context $\mathbf{x}$ and the correct answer $\mathbf{y}$. These assumptions characterize the problem of program synthesis well, but they also apply to many other discrete optimization and structured prediction domains.

**Simplified notation.** In what follows, we simplify the notation by dropping the dependence of the policy and the reward on $\mathbf{x}$ and $\mathbf{y}$. We use a notation of $\pi_\theta(\mathbf{a})$ instead of $\pi_\theta(\mathbf{a} \mid \mathbf{x})$ and $R(\mathbf{a})$ instead of $R(\mathbf{a} \mid \mathbf{x}, \mathbf{y})$, to make the formulation less cluttered, but the equations hold in the general case.

We express the expected return objective in the simplified notation as,

$$\mathcal{O}_{\mathrm{ER}}(\theta) = \sum_{\mathbf{a} \in \mathcal{A}} \pi_\theta(\mathbf{a}) R(\mathbf{a}) = \mathbb{E}_{\mathbf{a} \sim \pi_\theta(\mathbf{a})} R(\mathbf{a}) . \tag{1}$$

The REINFORCE [58] algorithm presents an elegant and convenient way to estimate the gradient of the expected return (1) using Monte Carlo (MC) samples. Using $K$ trajectories sampled *i.i.d.* from the current policy $\pi_\theta$, denoted $\{\mathbf{a}^{(1)}, \ldots, \mathbf{a}^{(K)}\}$, the gradient estimate can be expressed as,

$$\nabla_\theta \mathcal{O}_{\mathrm{ER}}(\theta) = \mathbb{E}_{\mathbf{a} \sim \pi_\theta(\mathbf{a})} \nabla \log \pi_\theta(\mathbf{a}) R(\mathbf{a}) \approx \frac{1}{K} \sum_{k=1}^K \nabla \log \pi_\theta(\mathbf{a}^{(k)}) \left[ R(\mathbf{a}^{(k)}) - b \right] , \tag{2}$$

where a baseline $b$ is subtracted from the returns to reduce the variance of gradient estimates. This formulation enables direct optimization of $\mathcal{O}_{\mathrm{ER}}$ via MC sampling from an unknown search space, which also serves the purpose of exploration. To improve such exploration behavior, one often includes the entropy of the policy as an additional term inside the objective to prevent early convergence. However, the key limitation of the formulation stems from the difficulty of estimating the gradients accurately only using a few *fresh* samples.

# 4 MAPO: Memory Augmented Policy Optimization

We consider an RL environment with a finite number of discrete actions, deterministic transitions, and deterministic terminal returns. In other words, the set of all possible action trajectories $\mathcal{A}$ is countable, even though possibly infinite, and re-evaluating the return of a trajectory $R(\mathbf{a})$ twice results in the same value. These assumptions characterize the problem of program synthesis well, but also apply to many structured prediction problems [47, 37] and combinatorial optimization domains (*e.g.*, [7]).

To reduce the variance in gradient estimation and prevent forgetting high-reward trajectories, we introduce a memory buffer, which saves a set of promising trajectories denoted $\mathcal{B} \equiv \{(\mathbf{a}^{(i)})\}_{i=1}^M$. Previous works [26, 2, 60] utilized a memory buffer by adopting a training objective similar to

$$\mathcal{O}_{\mathrm{AUG}}(\theta) = \lambda \mathcal{O}_{\mathrm{ER}}(\theta) + (1 - \lambda) \sum_{\mathbf{a} \in \mathcal{B}} \log \pi_\theta(\mathbf{a}), \tag{3}$$

which combines the expected return objective with a maximum likelihood objective over the memory buffer $\mathcal{B}$. This training objective is not directly optimizing the expected return any more because the second term introduces bias into the gradient. When the trajectories in $\mathcal{B}$ are not gold trajectories

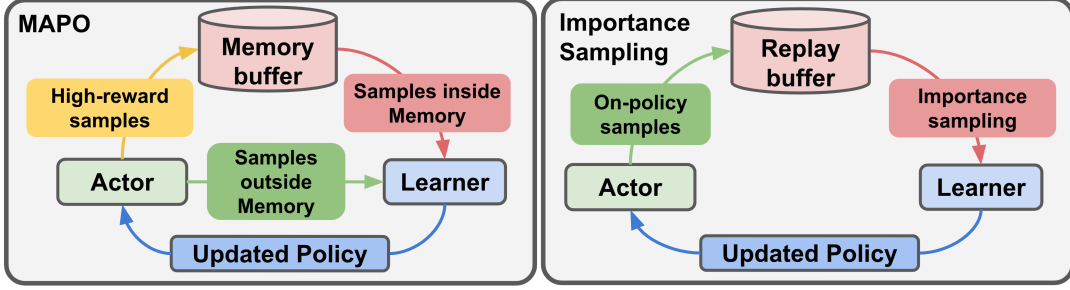

Figure 1: Overview of MAPO compared with experience replay using importance sampling.

but high-reward trajectories collected during exploration, uniformly maximizing the likelihood of each trajectory in $\mathcal{B}$ could be problematic. For example, in program synthesis, there can sometimes be spurious programs [40] that get the right answer, thus receiving high reward, for a wrong reason, e.g., using $2 + 2$ to answer the question "what is two times two". Maximizing the likelihood of those high-reward but spurious programs will bias the gradient during training.

We aim to utilize the memory buffer in a principled way. Our key insight is that one can re-express the expected return objective as a weighted sum of two terms: an expectation over the trajectories inside the memory buffer, and a separate expectation over the trajectories outside the buffer,

$$\mathcal{O}_{\mathrm{ER}}(\theta) = \sum_{\mathbf{a}\in\mathcal{B}} \pi_\theta(\mathbf{a})\, R(\mathbf{a}) \quad + \sum_{\mathbf{a}\in(\mathcal{A}-\mathcal{B})} \pi_\theta(\mathbf{a})\, R(\mathbf{a}) \tag{4}$$

$$= \pi_\mathcal{B} \underbrace{\mathbb{E}_{\mathbf{a}\sim\pi_\theta^+(\mathbf{a})} R(\mathbf{a})}_{\text{Expectation inside } \mathcal{B}} \quad + \quad (1-\pi_\mathcal{B}) \underbrace{\mathbb{E}_{\mathbf{a}\sim\pi_\theta^-(\mathbf{a})} R(\mathbf{a})}_{\text{Expectation outside } \mathcal{B}}, \tag{5}$$

where $\mathcal{A} - \mathcal{B}$ denotes the set of trajectories not included in the memory buffer, $\pi_\mathcal{B} = \sum_{\mathbf{a}\in\mathcal{B}} \pi_\theta(\mathbf{a})$ denote the total probability of the trajectories in the buffer, and $\pi_\theta^+(\mathbf{a})$ and $\pi_\theta^-(\mathbf{a})$ denote a normalized probability distribution inside and outside of the buffer,

$$\pi_\theta^+(\mathbf{a}) = \begin{cases} \pi_\theta(\mathbf{a})/\pi_\mathcal{B} & \text{if } \mathbf{a}\in\mathcal{B} \\ 0 & \text{if } \mathbf{a}\notin\mathcal{B} \end{cases}, \qquad \pi_\theta^-(\mathbf{a}) = \begin{cases} 0 & \text{if } \mathbf{a}\in\mathcal{B} \\ \pi_\theta(\mathbf{a})/(1-\pi_\mathcal{B}) & \text{if } \mathbf{a}\notin\mathcal{B} \end{cases}. \tag{6}$$

The policy gradient can be expressed as,

$$\nabla_\theta \mathcal{O}_{\mathrm{ER}}(\theta) = \pi_\mathcal{B}\, \mathbb{E}_{\mathbf{a}\sim\pi_\theta^+(\mathbf{a})} \nabla \log \pi_\theta(\mathbf{a}) R(\mathbf{a}) + (1-\pi_\mathcal{B})\, \mathbb{E}_{\mathbf{a}\sim\pi_\theta^-(\mathbf{a})} \nabla \log \pi_\theta(\mathbf{a}) R(\mathbf{a}). \tag{7}$$

The second expectation can be estimated by sampling from $\pi_\theta^-(\mathbf{a})$, which can be done through rejection sampling by sampling from $\pi_\theta(\mathbf{a})$ and rejecting the sample if $\mathbf{a}\in\mathcal{B}$. If the memory buffer only contains a small number of trajectories, the first expectation can be computed exactly by enumerating all the trajectories in the buffer. The variance in gradient estimation is reduced because we get an exact estimate of the first expectation while sampling from a smaller stochastic space of measure $(1-\pi_\mathcal{B})$. If the memory buffer contains a large number of trajectories, the first expectation can be approximated by sampling. Then, we get a *stratified sampling* estimator of the gradient. The trajectories inside and outside the memory buffer are two mutually exclusive and collectively exhaustive strata, and the variance reduction still holds. The weights for the first and second expectations are $\pi_\mathcal{B}$ and $1 - \pi_\mathcal{B}$ respectively. We call $\pi_\mathcal{B}$ the *memory weight*.

In the following we present 3 techniques to make an efficient algorithm of MAPO.

## 4.1 Memory Weight Clipping

Policy gradient methods usually suffer from a cold start problem. A key observation is that a "bad" policy, one that achieves low expected return, will assign small probabilities to the high-reward trajectories, which in turn causes them to be ignored during gradient estimation. So it is hard to improve from a random initialization, *i.e.*, the cold start problem, or to recover from a bad update, *i.e.*, the brittleness problem. Ideally we want to force the policy gradient estimates to pay at least some attention to the high-reward trajectories. Therefore, we adopt a clipping mechanism over the

memory weight $\pi_{\mathcal{B}}$, which ensures that the memory weight is greater or equal to $\alpha$, *i.e.*, $\pi_{\mathcal{B}} \geqslant \alpha$, otherwise clips it to $\alpha$. So the new gradient estimate is,

$$\nabla_\theta \mathcal{O}_{\mathrm{ER}}^c(\theta) = \pi_{\mathcal{B}}^c \, \mathbb{E}_{\mathbf{a} \sim \pi_\theta^+(\mathbf{a})} \nabla \log \pi_\theta(\mathbf{a}) R(\mathbf{a}) + (1 - \pi_{\mathcal{B}}^c) \, \mathbb{E}_{\mathbf{a} \sim \pi_\theta^-(\mathbf{a})} \nabla \log \pi_\theta(\mathbf{a}) R(\mathbf{a}), \quad (8)$$

where $\pi_{\mathcal{B}}^c = \max(\pi_{\mathcal{B}}, \alpha)$ is the clipped memory weight. At the beginning of training, the clipping is active and introduce a bias, but accelerates and stabilizes training. Once the policy is off the ground, the memory weights are almost never clipped given that they are naturally larger than $\alpha$ and the gradients are not biased any more. See section 5.4 for an empirical analysis of the clipping.

## 4.2 Systematic Exploration

To discover high-reward trajectories for the memory buffer $\mathcal{B}$, we need to efficiently explore the search space. Exploration using policy samples suffers from repeated samples, which is a waste of computation in deterministic environments. So we propose to use systematic exploration to improve the efficiency. More specifically we keep a set $\mathcal{B}^e$ of fully explored partial sequences, which can be efficiently implemented using a bloom filter. Then, we use it to enforce a policy to only take actions that lead to unexplored sequences. Using a bloom filter we can store billions of sequences in $\mathcal{B}^e$ with only several gigabytes of memory. The pseudo code of this approach is shown in Algorithm 1. We warm start the memory buffer using systematic exploration from random policy as it can be trivially parallelized. In parallel to training, we continue the systematic exploration with the current policy to discover new high reward trajectories.

## 4.3 Distributed Sampling

An exact computation of the first expectation of (5) requires an enumeration over the memory buffer. The cost of gradient computation will grow linearly *w.r.t* the number of trajectories in the buffer, so it can be prohibitively slow when the buffer contains a large number of trajectories. Alternatively, we can approximate the first expectation using sampling. As mentioned above, this can be viewed as *stratified sampling* and the variance is still reduced. Although the cost of gradient computation now grows linearly *w.r.t* the number of samples instead of the total number of trajectories in the buffer, the cost of sampling still grows linearly *w.r.t* the size of the memory buffer because we need to compute the probability of each trajectory with the current model.

A key insight is that if the bottleneck is in sampling, the cost can be distributed through an actor-learner architecture similar to [15]. See the Supplemental Material D for a figure depicting the actor-learner architecture. The actors each use its model to sample trajectories from inside the memory buffer through renormalization ($\pi_\theta^+$ in (6)), and uses rejection sampling to pick trajectories from outside the memory ($\pi_\theta^-$ in (6)). It also computes the weights for these trajectories using the model. These trajectories and their weights are then pushed to a queue of samples. The learner fetches samples from the queue and uses

---

**Algorithm 1** Systematic Exploration

**Input:** context $\mathbf{x}$, policy $\pi$, fully explored sub-sequences $\mathcal{B}^e$, high-reward sequences $\mathcal{B}$
**Initialize:** empty sequence $a_{0:0}$
**while** true **do**
$\quad V = \{a \mid a_{0:t-1} \| a \notin B^e\}$
$\quad$ **if** $V == \varnothing$ **then**
$\quad\quad \mathcal{B}^e \leftarrow \mathcal{B}^e \cup a_{0:t-1}$
$\quad\quad$ **break**
$\quad$ sample $a_t \sim \pi^V(a|a_{0:t-1})$
$\quad a_{0:t} \leftarrow a_{0:t-1} \| a_t$
$\quad$ **if** $a_t ==$ EOS **then**
$\quad\quad$ **if** $R(a_{0:t}) > 0$ **then**
$\quad\quad\quad \mathcal{B} \leftarrow \mathcal{B} \cup a_{0:t}$
$\quad\quad \mathcal{B}^e \leftarrow \mathcal{B}^e \cup a_{0:t}$
$\quad\quad$ **break**

---

**Algorithm 2** MAPO

**Input:** data $\{(\mathbf{x}_i, \mathbf{y}_i)\}_{i=1}^N$, memories $\{(\mathcal{B}_i, \mathcal{B}_i^e)\}_{i=1}^N$, constants $\alpha, \epsilon, M$
**repeat** $\qquad\qquad\quad \triangleright$ **for all actors**
$\quad$ Initialize training batch $D \leftarrow \varnothing$
$\quad$ Get a batch of inputs $C$
$\quad$ **for** $(\mathbf{x}_i, \mathbf{y}_i, \mathcal{B}_i^e, \mathcal{B}_i) \in C$ **do**
$\quad\quad$ Algorithm1$(\mathbf{x}_i, \pi_\theta^{old}, \mathcal{B}_i^e, \mathcal{B}_i)$
$\quad\quad$ Sample $\mathbf{a}_i^+ \sim \pi_\theta^{old}$ over $\mathcal{B}_i$
$\quad\quad w_i^+ \leftarrow \max(\pi_\theta^{old}(\mathcal{B}_i), \alpha)$
$\quad\quad D \leftarrow D \cup (\mathbf{a}_i^+, R(\mathbf{a}_i^+), w_i^+)$
$\quad\quad$ Sample $\mathbf{a}_i \sim \pi_\theta^{old}$
$\quad\quad$ **if** $\mathbf{a}_i \notin \mathcal{B}_i$ **then**
$\quad\quad\quad w_i \leftarrow (1 - w_i^+)$
$\quad\quad\quad D \leftarrow D \cup (\mathbf{a}_i, R(\mathbf{a}_i), w_i)$
$\quad$ Push $D$ to training queue
**until** converge or early stop
**repeat** $\qquad\qquad\quad \triangleright$ **for the learner**
$\quad$ Get a batch $D$ from training queue
$\quad$ **for** $(\mathbf{a}_i, R(\mathbf{a}_i), w_i) \in D$ **do**
$\quad\quad \mathrm{d}\theta \leftarrow \mathrm{d}\theta + w_i \, R(\mathbf{a}_i) \, \nabla \log \pi_\theta(\mathbf{a}_i)$
$\quad$ update $\theta$ using $\mathrm{d}\theta$
$\quad \pi_\theta^{old} \leftarrow \pi_\theta \quad \triangleright$ once every M batches
**until** converge or early stop
**Output:** final parameters $\theta$

---

them to compute gradient estimates to update the parameters. By distributing the cost of sampling to a set of actors, the training can be accelerated almost linearly *w.r.t* the number of actors. In our experiments, we got a $\sim$20 times speedup from distributed sampling with 30 actors.

## 4.4 Final Algorithm

The final training procedure is summarized in Algorithm 2. As mentioned above, we adopt the actor-learner architecture for distributed training. It uses multiple actors to collect training samples asynchronously and one learner for updating the parameters based on the training samples. Each actor interacts with a set of environments to generate new trajectories. For efficiency, an actor uses a stale policy ($\pi_\theta^{old}$), which is often a few steps behind the policy of the learner and will be synchronized periodically. To apply MAPO, each actor also maintains a memory buffer $\mathcal{B}_i$ to save the high-reward trajectories. To prepare training samples for the learner, the actor picks $n_b$ samples from inside $\mathcal{B}_i$ and also performs rejection sampling with $n_o$ on-policy samples, both according to the actor's policy $\pi_\theta^{old}$. We then use the actor policy to compute a weight $max(\pi_\theta(\mathcal{B}), \alpha)$ for the samples in the memory buffer, and use $1 - max(\pi_\theta(\mathcal{B}), \alpha)$ for samples outside of the buffer. These samples are pushed to a queue and the learner reads from the queue to compute gradients and update the parameters.

## 5 Experiments

We evaluate MAPO on two program synthesis from natural language (also known as *semantic parsing*) benchmarks, WIKITABLEQUESTIONS and WIKISQL, which requires generating programs to query and process data from tables to answer natural language questions. We first compare MAPO to four common baselines, and ablate systematic exploration and memory weight clipping to show their utility. Then we compare MAPO to the state-of-the-art on these two benchmarks. On WIKITABLEQUESTIONS, MAPO is the first RL-based approach that significantly outperforms the previous state-of-the-art. On WIKISQL, MAPO trained with weak supervision (question-answer pairs) outperforms several strong models trained with full supervision (question-program pairs).

### 5.1 Experimental setup

**Datasets.** WIKITABLEQUESTIONS [39] contains tables extracted from Wikipedia and question-answer pairs about the tables. See Table 1 as an example. There are 2,108 tables and 18,496 question-answer pairs splitted into train/dev/test set.. We follow the construction in [39] for converting a table into a directed graph that can be queried, where rows and cells are converted to graph nodes while column names become labeled directed edges. For the questions, we use string match to identify phrases that appear in the table. We also identify numbers and dates using the CoreNLP annotation released with the dataset. The task is challenging in several aspects. First, the tables are taken from Wikipedia and cover a wide range of topics. Second, at test time, new tables that contain unseen column names appear. Third, the table contents are not normalized as in knowledge-bases like Freebase, so there are noises and ambiguities in the table annotation. Last, the semantics are more complex comparing to previous datasets like WEBQUESTIONSSP [62]. It requires multiple-step reasoning using a large set of functions, including comparisons, superlatives, aggregations, and arithmetic operations [39]. See Supplementary Material A for more details about the functions.

WIKISQL [68] is a recent large scale dataset on learning natural language interfaces for databases. It also uses tables extracted from Wikipedia, but is much larger and is annotated with programs (SQL). There are 24,241 tables and 80,654 question-program pairs splitted into train/dev/test set. Comparing to WIKITABLEQUESTIONS, the semantics are simpler because the SQLs use fewer operators (column selection, aggregation, and conditions). We perform similar preprocessing as for WIKITABLEQUESTIONS. Most of the state-of-the-art models relies on question-program pairs for supervised training, while we only use the question-answer pairs for weakly supervised training.

**Model architecture.** We adopt the Neural Symbolic Machines framework[26], which combines (1) a neural "programmer", which is a seq2seq model augmented by a key-variable memory that can translate a natural language utterance to a program as a sequence of tokens, and (2) a symbolic "computer", which is an Lisp interpreter that implements a domain specific language with built-in functions and provides code assistance by eliminating syntactically or semantically invalid choices.

For the Lisp interpreter, we added functions according to [67, 34] for WIKITABLEQUESTIONS experiments and used the subset of functions equivalent to column selection, aggregation, and conditions for WIKISQL. See the Supplementary Material A for more details about functions used.

We implemented the seq2seq model augmented with key-variable memory from [26] in Tensor-Flow [1]. Some minor differences are: (1) we used a bi-directional LSTM for the encoder; (2) we used two-layer LSTM with skip-connections in both the encoder and decoder. GloVe [43] embeddings are used for the embedding layer in the encoder and also to create embeddings for column names by

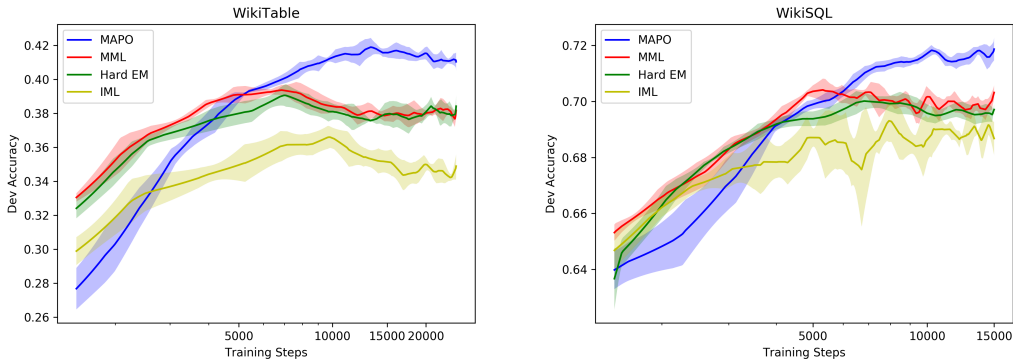

Figure 2: Comparison of MAPO and 3 baselines' dev set accuracy curves. Results on WIKITABLE-QUESTIONS is on the left and results on WIKISQL is on the right. The plot is average of 5 runs with a bar of one standard deviation. The horizontal coordinate (training steps) is in log scale.

averaging the embeddings of the words in a name. Following [34, 24], we also add a binary feature in each step of the encoder, indicating whether this word is found in the table, and an integer feature for a column name counting how many of the words in the column name appear in the question. For the WIKITABLEQUESTIONS dataset, we use the CoreNLP annotation of numbers and dates released with the dataset. For the WIKISQL dataset, only numbers are used, so we use a simple parser to identify and parse the numbers in the questions, and the tables are already preprocessed. The tokens of the numbers and dates are anonymized as two special tokens <NUM> and <DATE>. The hidden size of the LSTM is 200. We keep the GloVe embeddings fixed during training, but project it to 200 dimensions using a trainable linear transformation. The same architecture is used for both datasets.

**Training Details.** We first apply systematic exploration using a random policy to discover high-reward programs to warm start the memory buffer of each example. For WIKITABLEQUESTIONS, we generated 50k programs per example using systematic exploration with pruning rules inspired by the grammars from [67] (see Supplementary E). We apply 0.2 dropout on both encoder and decoder. Each batch includes samples from 25 examples. For experiments on WIKISQL, we generated 1k programs per example due to computational constraint. Because the dataset is much larger, we don't use any regularization. Each batch includes samples from 125 examples. We use distributed sampling for WIKITABLEQUESTIONS. For WIKISQL, due to computational constraints, we truncate each memory buffer to top 5 and then enumerate all 5 programs for training. For both experiments, the samples outside memory buffer are drawn using rejection sampling from 1 on-policy sample per example. At inference time, we apply beam search of size 5. We evaluate the model periodically on the dev set to select the best model. We apply a distributed actor-learner architecture for training. The actors use CPUs to generate new trajectories and push the samples into a queue. The learner reads batches of data from the queue and uses GPU to accelerate training (see Supplementary D). We use Adam optimizer for training and the learning rate is $10^{-3}$. All the hyperparameters are tuned on the dev set. We train the model for 25k steps on WikiTableQuestions and 15k steps on WikiSQL.

## 5.2 Comparison to baselines

We first compare MAPO against the following baselines using the same neural architecture.

▸ **REINFORCE:** We use on-policy samples to estimate the gradient of expected return as in (2), not utilizing any form of memory buffer.

▸ **MML:** Maximum Marginal Likelihood maximizes the marginal probability of the memory buffer as in $\mathcal{O}_{\text{MML}}(\theta) = \frac{1}{N} \sum_i \log \sum_{\mathbf{a} \in \mathcal{B}_i} \pi_\theta(\mathbf{a}) = \frac{1}{N} \log \prod_i \sum_{\mathbf{a} \in \mathcal{B}_i} \pi_\theta(\mathbf{a})$. Assuming binary rewards and assuming that the memory buffer contains almost all of the trajectories with a reward of 1, MML optimizes the marginal probability of generating a rewarding program. Note that under these assumptions, expected return can be expressed as $\mathcal{O}_{\text{ER}}(\theta) \approx \frac{1}{N} \sum_i \sum_{\mathbf{a} \in \mathcal{B}_i} \pi_\theta(\mathbf{a})$. Comparing the two objectives, we can see that MML maximizes the product of marginal probabilities, whereas expected return maximizes the sum. More discussion of these two objectives can be found in [17, 36, 48].

▸ **Hard EM:** Expectation-Maximization algorithm is commonly used to optimize the marginal likelihood in the presence of latent variables. Hard EM uses the samples with the highest probability

to approximate the gradient to $\mathcal{O}_{\mathrm{MML}}$.

▸ **IML:** Iterative Maximum Likelihood training [26, 2] uniformly maximizes the likelihood of all the trajectories with the highest rewards $\mathcal{O}_{\mathrm{ML}}(\theta) = \sum_{\mathbf{a} \in \mathcal{B}} \log \pi_\theta(\mathbf{a})$.

Because the memory buffer is too large to enumerate, we use samples from the buffer to approximate the gradient for MML and IML, and uses samples with highest $\pi_\theta(\mathbf{a})$ for Hard EM.

We show the result in Table 2 and the dev accuracy curves in Figure 2. Removing systematic exploration or the memory weight clipping significantly weaken MAPO because high-reward trajectories are not found or easily forgotten. REINFORCE barely learns anything because starting from a random policy, most samples result in a reward of zero. MML and Hard EM converge faster, but the learned models underperform MAPO, which suggests that the expected return is a better objective. IML runs faster because it randomly samples from the buffer, but the objective is prone to spurious programs.

### 5.3 Comparison to state-of-the-art

On WIKITABLEQUESTIONS (Table 3), MAPO is the first RL-based approach that significantly outperforms the previous state-of-the-art by 2.6%. Unlike previous work, MAPO does not require manual feature engineering or additional human annotation[1]. On WIKISQL (Table 4), even though MAPO does not exploit ground truth programs (weak supervision), it is able to outperform many strong baselines trained using programs (full supervision). The techniques introduced in other models can be incorporated to further improve the result of MAPO, but we leave that as future work. We also qualitatively analyzed a trained model and see that it can generate fairly complex programs. See the Supplementary Material B for some examples of generated programs. We select the best model based on validation accuracy and report its test accuracy. We also report the mean accuracy and standard deviation based on 5 runs given the variance caused by the nonlinear optimization procedure, although it is not available from other models.

### 5.4 Analysis of Memory Weight Clipping

In this subsection, we present an analysis of the bias introduced by memory weight clipping. We define the clipping fraction as the percentage of examples where the clipping is active. In other words, it is the percentage of examples with a non-empty memory buffer, for which $\pi_{\mathcal{B}} < \alpha$. It is also the fraction of examples whose gradient computation will be biased by the clipping, so the higher the value, the more bias, and the gradient is unbiased when the clip fraction is zero. In figure 3, one can observe that the clipping fraction approaches zero towards the end of training and is negatively correlated with the training accuracy. In the experiments, we found that a fixed clipping threshold works well, but we can also gradually decrease the clipping threshold to completely remove the bias.

|  | WIKITABLE | WIKISQL |
|---|---|---|
| REINFORCE | $< 10$ | $< 10$ |
| MML (Soft EM) | $39.7 \pm 0.3$ | $70.7 \pm 0.1$ |
| Hard EM | $39.3 \pm 0.6$ | $70.2 \pm 0.3$ |
| IML | $36.8 \pm 0.5$ | $70.1 \pm 0.2$ |
| MAPO | $\mathbf{42.3} \pm 0.3$ | $\mathbf{72.2} \pm 0.2$ |
| MAPO w/o SE | $< 10$ | $< 10$ |
| MAPO w/o MWC | $< 10$ | $< 10$ |

Table 2: Ablation study for Systematic Exploration (SE) and Memory Weight Clipping (MWC). We report mean accuracy %, and its standard deviation on dev sets based on 5 runs.

|  | E.S. | Dev. | Test |
|---|---|---|---|
| Pasupat & Liang (2015) [39] | - | 37.0 | 37.1 |
| Neelakantan *et al.* (2017) [34] | 1 | 34.1 | 34.2 |
| Neelakantan *et al.* (2017) [34] | 15 | 37.5 | 37.7 |
| Haug *et al.* (2017) [18] | 1 | - | 34.8 |
| Haug *et al.* (2017) [18] | 15 | - | 38.7 |
| Zhang *et al.* (2017) [67] | - | 40.4 | 43.7 |
| MAPO | 1 | 42.7 | 43.8 |
| MAPO (mean of 5 runs) | - | 42.3 | 43.1 |
| MAPO (std of 5 runs) | - | 0.3 | 0.5 |
| MAPO (ensembled) | 10 | - | **46.3** |

Table 3: Results on WIKITABLEQUESTIONS. E.S. is the ensemble size, when applicable.

| **Fully supervised** | Dev. | Test |
|---|---|---|
| Zhong *et al.* (2017) [68] | 60.8 | 59.4 |
| Wang *et al.* (2017) [56] | 67.1 | 66.8 |
| Xu *et al.* (2017) [61] | 69.8 | 68.0 |
| Huang *et al.* (2018) [22] | 68.3 | 68.0 |
| Yu *et al.* (2018) [63] | 74.5 | 73.5 |
| Sun *et al.* (2018) [54] | 75.1 | 74.6 |
| Dong & Lapata (2018) [14] | **79.0** | **78.5** |
| **Weakly supervised** | Dev. | Test |
| MAPO | **72.2** | 72.6 |
| MAPO (mean of 5 runs) | 72.2 | 72.1 |
| MAPO (std of 5 runs) | 0.2 | 0.3 |
| MAPO (ensemble of 10) | - | **74.9** |

Table 4: Results on WIKISQL. Unlike other methods, MAPO only uses weak supervision.

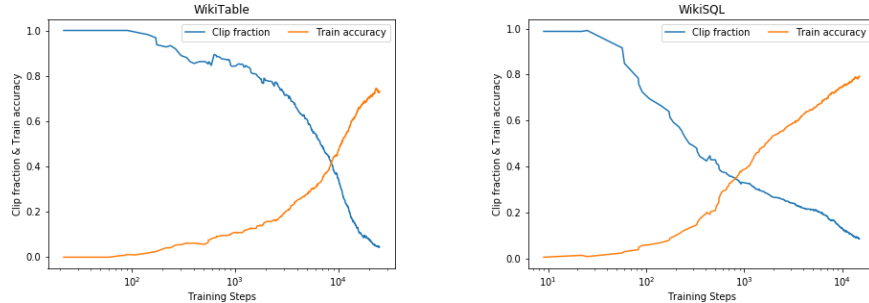

Figure 3: The clipping fraction and training accuracy w.r.t the training steps (log scale).

## 6    Related work

**Program synthesis & semantic parsing.** There has been a surge of recent interest in applying reinforcement learning to program synthesis [10, 2, 64, 33] and combinatorial optimization [70, 7]. Different from these efforts, we focus on the contextualized program synthesis where generalization to new contexts is important. Semantic parsing [65, 66, 27] maps natural language to executable symbolic representations. Training semantic parsers through weak supervision is challenging because the model must interact with a symbolic interpreter through non-differentiable operations to search over a large space of programs [8, 26]. Previous work [17, 34] reports negative results when applying simple policy gradient methods like REINFORCE [58], which highlights the difficulty of exploration and optimization when applying RL techniques. MAPO takes advantage of discrete and deterministic nature of program synthesis and significantly improves upon REINFORCE.

**Experience replay.** An experience replay buffer [28] enables storage and usage of past experiences to improve the sample efficiency of RL algorithms. Prioritized experience replay [49] prioritizes replays based on temporal-difference error for more efficient optimization. Hindsight experience replay [4] incorporates goals into replays to deal with sparse rewards. MAPO also uses past experiences to tackle sparse reward problems, but by storing and reusing high-reward trajectories, similar to [26, 38]. Previous work[26] assigns a fixed weight to the trajectories, which introduces bias into the policy gradient estimates. More importantly, the policy is often trained equally on the trajectories that have the same reward, which is prone to spurious programs. By contrast, MAPO uses the trajectories in a principled way to obtain an unbiased low variance gradient estimate.

**Variance reduction.** Policy optimization via gradient descent is challenging because of: (1) large *variance* in gradient estimates; (2) small gradients in the initial phase of training. Prior variance reduction approaches [59, 58, 29, 16] mainly relied on control variate techniques by introducing a critic model [23, 31, 51]. MAPO takes a different approach to reformulate the gradient as a combination of expectations inside and outside a memory buffer. Standard solutions to the small gradient problem involves supervised pretraining [52, 19, 46] or using supervised data to generate rewarding samples [36, 13], which cannot be applied when supervised data are not available. MAPO reduces the variance by sampling from a smaller stochastic space or through stratified sampling, and accelerates and stabilizes training by clipping the weight of the memory buffer.

**Exploration.** Recently there has been a lot of work on improving exploration [42, 55, 21] by introducing additional reward based on information gain or pseudo count. For program synthesis [5, 34, 10], the search spaces are enumerable and deterministic. Therefore, we propose to conduct systematic exploration, which ensures that only novel trajectories are generated.

## 7    Conclusion

We present memory augmented policy optimization (MAPO) that incorporates a memory buffer of promising trajectories to reduce the variance of policy gradients. We propose 3 techniques to enable an efficient algorithm for MAPO: (1) memory weight clipping to accelerate and stabilize training; (2) systematic exploration to efficiently discover high-reward trajectories; (3) distributed sampling from inside and outside memory buffer to scale up training. MAPO is evaluated on real world program synthesis from natural language / semantic parsing tasks. On WIKITABLEQUESTIONS, MAPO is the first RL approach that significantly outperforms previous state-of-the-art; on WIKISQL, MAPO trained with only weak supervision outperforms several strong baselines trained with full supervision.

**Acknowledgments**

We would like to thank Dan Abolafia, Ankur Taly, Thanapon Noraset, Arvind Neelakantan, Wenyun Zuo, Chenchen Pan and Mia Liang for helpful discussions. Jonathan Berant was partially supported by The Israel Science Foundation grant 942/16.

## Footnotes

[1] Krishnamurthy *et al.* [24] achieved 45.9 accuracy when trained on the data collected with dynamic programming and pruned with more human annotations [41, 32].

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
