[Supplementary Material]

# A  Domain Specific Language

We adopt a Lisp-like domain specific language with certain built-in functions. A program $C$ is a list of expressions $(c_1...c_N)$, where each expression is either a special token "*EOS*" indicating the end of the program, or a list of tokens enclosed by parentheses "$(F A_1...A_K)$". $F$ is a function, which takes as input $K$ arguments of specific types. Table A defines the arguments, return value and semantics of each function. In the table domain, there are rows and columns. The value of the table cells can be number, date time or string, so we also categorize the columns into number columns, date time columns and string columns depending on the type of the cell values in the column.

| Function | Arguments | Returns | Description |
|---|---|---|---|
| (**hop** v p) | **v**: a list of rows.<br>**p**: a column. | a list of cells. | Select the given column of the given rows. |
| (**argmax** v p)<br>(**argmin** v p) | **v**: a list of rows.<br>**p**: a number or date column. | a list of rows. | From the given rows, select the ones with the largest / smallest value in the given column. |
| (**filter**$_>$ v q p)<br>(**filter**$_\geqslant$ v q p)<br>(**filter**$_<$ v q p)<br>(**filter**$_\leqslant$ v q p)<br>(**filter**$_=$ v q p)<br>(**filter**$_{\neq}$ v q p) | **v**: a list of rows.<br>**q**: a number or date.<br>**p**: a number or date column. | a list of rows. | From the given rows, select the ones whose given column has certain order relation with the given value. |
| (**filter**$_{in}$ v q p)<br>(**filter**$_{!in}$ v q p) | **v**: a list of rows.<br>**q**: a string.<br>**p**: a string column. | a list of rows. | From the given rows, select the ones whose given column contain / do not contain the given string. |
| (**first** v)<br>(**last** v) | **v**: a list of rows. | a row. | From the given rows, select the one with the smallest / largest index. |
| (**previous** v)<br>(**next** v) | **v**: a row. | a row. | Select the row that is above / below the given row. |
| (**count** v) | **v**: a list of rows. | a number. | Count the number of given rows. |
| (**max** v p)<br>(**min** v p)<br>(**average** v p)<br>(**sum** v p) | **v**: a list of rows.<br>**p**: a number column. | a number. | Compute the maximum / minimum / average / sum of the given column in the given rows. |
| (**mode** v p) | **v**: a list of rows.<br>**p**: a column. | a cell. | Get the most common value of the given column in the given rows. |
| (**same_as** v p) | **v**: a row.<br>**p**: a column. | a list of rows. | Get the rows whose given column is the same as the given row. |
| (**diff** v0 v1 p) | **v0**: a row.<br>**v1**: a row.<br>**p**: a number column. | a number. | Compute the difference in the given column of the given two rows. |

Table 5: Functions used in the experiments.

In the WIKITABLEQUESTIONS experiments, we used all the functions in the table. In the WIKISQL experiments, because the semantics of the questions are simpler, we used a subset of the functions (hop, filter$_=$, filter$_{in}$, filter$_>$, filter$_<$, count, maximum, minimum, average and sum). We created the functions according to [67, 34].[2]

## B Examples of Generated Programs

The following table shows examples of several types of programs generated by a trained model.

| Statement | Comment |
|---|---|
| **Superlative** | |
| **nt-13901: the most points were scored by which player?** | |
| (argmax all_rows r.points-num) | Sort all rows by column 'points' and take the first row. |
| (hop v0 r.player-str) | Output the value of column 'player' for the rows in v0. |
| **Difference** | |
| **nt-457: how many more passengers flew to los angeles than to saskatoon?** | |
| (filter$_{in}$ all_rows ['saskatoon'] r.city-str) | Find the row with 'saskatoon' matched in column 'city'. |
| (filter$_{in}$ all_rows ['los angeles'] r.city-str) | Find the row with 'los angeles' matched in column 'city'. |
| (diff v1 v0 r.passengers-num) | Calculate the difference of the values in the column 'passenger' of v0 and v1. |
| **Before / After** | |
| **nt-10832: which nation is before peru?** | |
| (filter$_{in}$ all_rows ['peru'] r.nation-str) | Find the row with 'peru' matched in 'nation' column. |
| (previous v0) | Find the row before v0. |
| (hop v1 r.nation-str) | Output the value of column 'nation' of v1. |
| **Compare & Count** | |
| **nt-647: in how many games did sri lanka score at least 2 goals?** | |
| (filter$_{\geqslant}$ all_rows [2] r.score-num) | Select the rows whose value in the 'score' column $\geq 2$. |
| (count v0) | Count the number of rows in v0. |
| **Exclusion** | |
| **nt-1133: other than william stuart price, which other businessman was born in tulsa?** | |
| (filter$_{in}$ all_rows ['tulsa'] r.hometown-str) | Find rows with 'tulsa' matched in column 'hometown'. |
| (filter$_{!in}$ v0 ['william stuart price'] r.name-str) | Drop rows with 'william stuart price' matched in the value of column 'name'. |
| (hop v1 r.name-str) | Output the value of column 'name' of v1. |

Table 6: Example programs generated by a trained model.

## C Analysis of Sampling from Inside and Outside Memory Buffer

In the following we give theoretical analysis of the distributed sampling approaches. For the purpose of the analysis we assume binary rewards, and exhaustive exploration, that the buffer $\mathcal{B}^+ \equiv \mathcal{B}$ contains all the high reward samples, and $\mathcal{B}^- \equiv \mathcal{A} - \mathcal{B}_a$ contains all the non-rewarded samples. It provides a general guidance of how examples should be allocated on the experiences and whether to use baseline or not so that the variance of gradient estimations can be minimized. In our work, we take the simpler approach to not use baseline and leave the empirical investigation to future work.

### C.1 Variance: baseline vs no baseline

Here we compare baseline strategies based on their variances of gradient estimations. We thank Wenyun Zuo's suggestion in approximating the variances.

Assume $\sigma_+^2 = Var_{\mathbf{a} \sim \pi_\theta^+(\mathbf{a})}[\nabla \log \pi(\mathbf{a})]$ and $\sigma_-^2 = Var_{\mathbf{a} \sim \pi_\theta^-(\mathbf{a})}[\nabla \log \pi(\mathbf{a})]$ are the variance of the gradient of the log likelihood inside and outside the buffer. If we don't use a baseline, the the optimal sampling strategy is to only sample from $\mathcal{B}$. The variance of the gradient estimation is

$$\mathrm{Var}[\nabla \mathcal{O}_{\mathrm{ER}}] \approx \pi_\theta(\mathcal{B})^2 \sigma_+^2 \tag{9}$$

If we use a baseline $b = \pi_\theta(\mathcal{B})$ and apply the optimal sampling allocation (Section C.2), then the variance of the gradient estimation is

$$\mathrm{Var}[\nabla \mathcal{O}_{\mathrm{ER}}]_b \approx \pi_\theta(\mathcal{B})^2(1 - \pi_\theta(\mathcal{B}))^2(\sigma_+^2 + \sigma_-^2) \tag{10}$$

We can prove that both of these estimators reduce the variance for the gradient estimation. To compare the two, we can see that the ratio of the variance with and without baseline is

$$\frac{\mathrm{Var}[\nabla \mathcal{O}_{\mathrm{ER}}]_b}{\mathrm{Var}[\nabla \mathcal{O}_{\mathrm{ER}}]} = (1 + \frac{\sigma_-^2}{\sigma_+^2})(1 - \pi_\theta(\mathcal{B}))^2 \tag{11}$$

So using baseline provides lower variance when $\pi_\theta(\mathcal{B}) \approx 1.0$, which roughly corresponds to the later stage of training, and when $\sigma_-$ is not much larger than $\sigma_+$; it is better to not use baselines when $\pi_\theta(\mathcal{B})$ is not close to 1.0 or when $\sigma_-$ is much larger than $\sigma_+$.

### C.2 Optimal Sample Allocation

Given that we want to apply stratified sampling to estimate the gradient of REINFORCE with baseline 2, here we show that the optimal sampling strategy is to allocate the same number of samples to $\mathcal{B}$ and $\mathcal{A} - \mathcal{B}$.

Assume that the gradient of log likelihood has similar variance on $\mathcal{B}$ and $\mathcal{A} - \mathcal{B}$:

$$\mathrm{Var}_{\pi_\theta^+(\mathbf{a})}[\nabla \log \pi_\theta(\mathbf{a})] \approx \mathrm{Var}_{\pi_\theta^-(\mathbf{a})}[\nabla \log \pi_\theta(\mathbf{a})] = \sigma^2 \tag{12}$$

The variance of gradient estimation on $\mathcal{B}$ and $\mathcal{A} - \mathcal{B}$ are:

$$\begin{aligned}
\mathrm{Var}_{\pi_\theta^+(\mathbf{a})}[(1 - \pi_\theta(\mathcal{B}))\nabla \log \pi_\theta(\mathbf{a})] &= (1 - \pi_\theta(\mathcal{B}))^2 * \sigma^2 \\
\mathrm{Var}_{\pi_\theta^-(\mathbf{a})}[(-\pi_\theta(\mathcal{B}))\nabla \log \pi_\theta(\mathbf{a})] &= \pi_\theta(\mathcal{B})^2 * \sigma^2
\end{aligned} \tag{13}$$

When performing stratified sampling, the optimal sample allocation is to let the number of samples from a stratum be proportional to its probability mass times the standard deviation $P_i \sigma_i$ In other words, the more probability mass and the more variance a stratum has, the more samples we should draw from a stratum. So the ratio of the number of samples allocated to each stratum under the optimal sample allocation is

$$\frac{k^+}{k^-} = \frac{\pi_\theta(\mathcal{B})\sqrt{\mathrm{Var}_{\pi_\theta^+(\mathbf{a})}}}{1 - \pi_\theta(\mathcal{B})\sqrt{\mathrm{Var}_{\pi_\theta^-(\mathbf{a})}}} \tag{14}$$

Using equation 13, we can see that

$$\frac{k^+}{k^-} = 1 \tag{15}$$

So the optimal strategy is to allocate the same number of samples to $\mathcal{B}$ and $\mathcal{A} - \mathcal{B}$.

## D Distributed Actor-Learner Architecture

Figure 4: Distributed actor-learner architecture.

Using 30 CPUs, each running one actor, and 2 GPUs, one for training and one for evaluating on dev set, the experiment finishes in about 3 hours on WikiTableQuestions and about 7 hours on WikiSQL.

# E  Pruning Rules for Random Exploration on WikiTableQuestions

The pruning rules are inspired by the grammar [67]. It can be seen as trigger words or POS tags for a subset of the functions. For the functions included, they are only allowed when at least one of the trigger words / tags appears in the sentence. For the other functions that are not included, there isn't any constraints. Also note that these rules are only used during random exploration. During training and evaluation, the rules are not applied.

| Function | Triggers |
|---|---|
| **count** | how, many, total, number |
| **filter**$_{!in}$ | not, other, besides |
| **first** | first, top |
| **last** | last, bottom |
| **argmin** | JJR, JJS, RBR, RBS, top, first, bottom, last |
| **argmax** | JJR, JJS, RBR, RBS, top, first, bottom, last |
| **sum** | all, combine, total |
| **average** | average |
| **maximum** | JJR, JJS, RBR, RBS |
| **minimum** | JJR, JJS, RBR, RBS |
| **mode** | most |
| **previous** | next, previous, after, before, above, below |
| **next** | next, previous, after, before, above, below |
| **same** | same |
| **diff** | difference, more, than |
| **filter**$_{\geqslant}$ | RBR, JJR, more, than, least, above, after |
| **filter**$_{\leqslant}$ | RBR, JJR, less, than, most, below, before, under |
| **filter**$_{>}$ | RBR, JJR, more, than, least, above, after |
| **filter**$_{<}$ | RBR, JJR, less, than, most, below, before, under |

Table 7: Pruning rules used during random exploration on WikiTableQuestions.

## Footnotes

[2]The only function we have added to capture some complex semantics is the same_as function, but it only appears in 1.2% of the generated programs (among which 0.6% are correct and the other 0.6% are incorrect), so even if we remove it, the significance of the difference in Table 3 will not change.