[Reviews · NeurIPS 2018]

Reviewer 1



This paper describes a Reinforcement Learning algorithm adapted to settings with sparse reward and weak supervision, and applies it to program synthesis, achieving state-of-the-art and even outperforming baselines with full supervision. The two first sections explain very clearly the motivation of this work, presenting the current limitations of reinforcement learning for tasks like contextual program synthesis. It is nicely written and pleasant to read. Section 3 presents the Reinforcement Learning framework that is the basis of the proposal, where the goal is to find a food approximation of the expected return objective. Section 4 presents the MAPO algorithm and his three key points: "(1) distributed sampling from inside and outside memory with an actor-learner architecture; (2) a marginal likelihood constraint over the memory to accelerate training; (3) systematic exploration to discover new high reward trajectories" (I did not find a better phrasing to summarize than the one in the abstract and the conclusion). The experiments are carried out on two public datasets, includes details for reproducibility, and show compelling results. The references are relevant throughout the paper. I cannot attest the completeness, not being an expert in RL. I don't see a reason why this paper should not be accepted! Some comments: * Probably a mistake on end of line 152: "smaller stochastic space of size xxx ..." -> "... of measure xxx ...". * Section 4.1 could be better explained. Why is the enumeration prohibitive? Because of the cost of evaluating new gradients and log probabilities for those trajectories? * End of section 4.2: "Once the policy is off the ground, almost never the buffer probabilities are truncated given that they are naturally larger than alpha". It would be interesting to see plots of this. no experimental results are given about any properties of these trajectories, but it seems like they would be important to look at to understand the technique. For example, what fraction of the gradient magnitude comes from the buffer and what fraction from the MC samples? Illustrating this would be the cherry on the cake.

Reviewer 2



This paper presents a new approach (MAPO) for policy gradient optimization in discrete and deterministic environments such as program synthesis (translating from natural language question to an SQL query). The main contribution is introducing a concept of policy memory that stores previously encountered high reward trajectories that is paired with a distributed sampling mechanism, a mechanism biasing initial sampling to high reward trajectories and systematic exploration of the trajectory space. For someone who is unfamiliar with the specific field, the paper is a bit challenging to follow. The reader must have some familiarity with policy gradient methods and the state of the art to fully appreciate the authors’ contributions. To evaluate their approach, the authors implement (with some modifications) a Neural Symbolic Machines architecture (Laing, ACL, 2017). This architecture first uses a sequences to sequences network to translate a natural langue question to a sequence of tokens that are converted to a valid SQL query using a List interpreter. Two datasets are used for testing: WikiSQL and WikiQuestions. The proposed approach generally outperforms accepted baselines. The general trend is that MAPO learns slower early in the process but achieves higher overall accuracy in the end. MAPO also outperforms state of the art approaches on both datasets by a non-trivial margin. Overall the paper is very well written (albeit a bit dense as a lot is covered from introduction of policy gradient, author’s innovations and specific NN implementation of the models). The field of automatic program synthesis is an important one and the proposed ideas can be applied to other domains exhibit similar phenomenology. Lastly, the proposed algorithm is demonstrated to outperform state of the art models on the benchmark datasets.

Reviewer 3



This paper proposes a modified form of policy gradient which can be used for RL tasks with deterministic dynamics, discrete states, and discrete actions. The basic idea is to combine gradients which marginalize over past behaviour using the current policy's probabilities, and gradients based on the usual stochastic roll-out-based approach. This produces an unbiased gradient estimator with reduced variance relative to using only stochastic roll-outs (e.g. plain REINFORCE). A couple tricks, which introduce some bias, are added to make the model perform well. The model is applied to the task of inferring programs from natural language specifications, and generally outperforms previous methods. The technical novelty of the method isn't great, but it seems to perform well when the required assumptions are met. The method is presented clearly, in a way that doesn't seem to exaggerate its technical sophistication. I appreciate the straightforward approach. The paper would be stronger if it included experiments on a reasonably distinct domain, so that we can have some confidence that the method's benefits are more broadly applicable. --- I have read the author rebuttal.